# RNA N6-Methyladenosine Modification in DNA Damage Response and Cancer Radiotherapy

**DOI:** 10.3390/ijms25052597

**Published:** 2024-02-23

**Authors:** Cui Wang, Shibo Yao, Tinghui Zhang, Xiaoya Sun, Chenjun Bai, Pingkun Zhou

**Affiliations:** 1College of Public Health, Hengyang Medical School, University of South China, Hengyang 421001, China; wangcui202102@163.com (C.W.); 2022000142@usc.edu.cn (X.S.); 2Beijing Key Laboratory for Radiobiology, Department of Radiation Biology, Beijing Institute of Radiation Medicine, Beijing 100850, China; yaoshibo@aliyun.com (S.Y.); 2053997571@aliyun.com (T.Z.)

**Keywords:** N6-methyladenosine, DNA damage response, radioresistance, cancer radiotherapy

## Abstract

The N6-methyladenosine (M6A) modification is the most common internal chemical modification of RNA molecules in eukaryotes. This modification can affect mRNA metabolism, regulate RNA transcription, nuclear export, splicing, degradation, and translation, and significantly impact various aspects of physiology and pathobiology. Radiotherapy is the most common method of tumor treatment. Different intrinsic cellular mechanisms affect the response of cells to ionizing radiation (IR) and the effectiveness of cancer radiotherapy. In this review, we summarize and discuss recent advances in understanding the roles and mechanisms of RNA M6A methylation in cellular responses to radiation-induced DNA damage and in determining the outcomes of cancer radiotherapy. Insights into RNA M6A methylation in radiation biology may facilitate the improvement of therapeutic strategies for cancer radiotherapy and radioprotection of normal tissues.

## 1. Introduction

RNA posttranscriptional modification (PTM) is an essential enzymatic processing of RNA molecules that influences RNAs’ functions in multiple aspects. Generally, RNA PTMs regulate the formation of R-loop to control transcription and the interactions between transcripts and trans-acting factors or other factors, such as RNA-binding proteins, to determine RNA functions, including stability, splicing, nuclear export, translation activity, etc. (Figure 1). Until now, more than 150 forms of RNA modifications have been detected [1]. N6-methyladenosine (M6A), a form of methylation occurring at the sixth N atom of the adenine base (A) of an RNA molecule, is currently considered to be the most abundant and conserved internal RNA modification. RNA M6A modification plays an essential role in RNA splicing, stability, output, degradation, and other metabolic processes [2], which affects almost all biological processes, such as cell autophagy [3], cell differentiation [4], inflammatory response [5], immune response [6], metabolic disease [7], carcinogenesis [8], and cancer prognosis [9].

The molecular compositions of the RNA M6A modification working system include the “writers” (adenosine methyltransferases), “readers” (RNA-binding proteins), and “erasers” (demethylases) (Figure 1). The M6A methyltransferase is a multicomponent complex composed of methyltransferase-like 3 (METTL3), methyltransferase-like 14 (METTL14), Wilms Protease 1-Protein (WTAP), vir like m6A methyltransferase associated (VIRMA, KIAA1429), RNA-binding motif protein 15/15B (RBM15/15B), and methyltransferase like 16 (METTL16) [10]. M6A modification is a dynamic and reversible process; it can be reversed by M6A demethylase. Fat mass and obesity-associated protein (FTO) is the first demethylase identified, affecting mRNA stability through oxidative reactions with substrates that lead to demethylase [11]. AlkB homolog 5, RNA demethylase (ALKBH5) is another demethylase that affects RNA metabolism; it differs from FTO in that ALKBH5 catalyzes the direct removal of methyl from M6A-methylated adenosine instead of oxidative demethylation. M6A modification works in two main ways mediated by the “reader” proteins: one way is by blocking or inducing protein–RNA binding through methylation and demethylation and the other is by recognizing the proteins or RNA by M6A-modified reading proteins, which cause subsequent reactions [12]. M6A readers consist of the YT521-B Homology (YTH) Domain family (YTHDF1/2/3), YTH Domain-containing proteins (YTHDC1/2), heterogeneous nuclear ribonucleoprotein (HNRNP) protein families, eukaryotic translation initiation factor 3 (eIF3), and insulin-like growth factor-2 mRNA-binding proteins 1/2/3 (IGF2BP1/2/3).

Radiotherapy is a common countermeasure for treating a wide range of tumors. Over the past 100 years, the knowledge and understanding regarding the biological response of various cells and tissues to ionizing radiation have been accumulating continuously. The advances in radiation biology have greatly improved outcomes and survival rates and reduced side effects of radiotherapy in cancer patients [13,14,15,16]. However, the mechanisms for the sensitive otherness and variation of cancer responses to radiation have not been fully explored. RNA M6A modification is a novel aspect of molecular processing, which can also determine the sensitivity of cells to radiation [17]. Undoubtedly, M6A modification plays a critical role in cellular responses to radiation, which could contribute to radioresistance, or vice versa, of cancers [18]. In this review, we summarize and discuss the recent research progress on RNA M6A modification in the field of radiation biology and the related significance in cancer radiotherapy.

## 2. The Role of M6A Modification in RNA Metabolism and Function

M6A methylation affects multiple processes of RNA metabolisms and functions, including pre-mRNA splicing, nuclear output, mRNA stability and translation, microRNA processing, LncRNA structure or function, and circRNA degradation and translation. After transcription, these steps of RNA metabolism have a significant impact on the level of gene expression.

### 2.1. Regulation of Pre-mRNA Splicing and RNA Nuclear Export and Effect on mRNA Stability

The splicing of the precursor of mRNA (hnRNA) is to remove the intron and the connection of the exon to create mature mRNA. M6A controls the splicing of hnRNA in two ways: directly and indirectly. In a direct mechanism, YTHDC1 recognizes M6A-modified hnRNA and recruits splicing factor 3 (SRSF3) to hnRNA, while it inhibits the binding of splicing factor recruits splicing factor 10 (SRSF10) to drive exon hopping, thereby regulating selective splicing of hnRNA [19]. The indirect approach is known as the “M6A switching mechanism”, in which the modification of M6A results in a change in the folded conformation of RNA, exposing the binding motif of HNRNPC, ultimately leading to the recruitment of HNRNPC and exon retention. For example, HNRNPC interacts with the M6A sites of TATA box-binding protein associated factor 8 (TAF8) pre-mRNA to promote exon skipping of TAF8, resulting in upregulation of the pro-metastasis isoform TAF8S [20]. In addition, M6A-binding proteins play an important role in regulating the nuclear output of RNA [21]. After being recognized by YTHDC1, the M6A-modified mRNA can interact with the export protein SRSF3 and is transmitted to the output receptor nuclear transcription factor, X-box binding 1 (NFX1) to form a complex, ultimately promoting the nuclear output of M6A-modified mRNAs [22].

Many studies have revealed that M6A modification is involved in the regulation of RNA stability. YTHDF2 recognizes translatable M6A-modified mRNA and introduces it into M6A decay sites, recruiting glucose-repressible alcohol dehydrogenase transcriptional effector (CCR4-NOT) deadenylase complexes to trigger the de-energization and degradation of transcripts [23]. SUMOylation (simulation of urban mobility) of YTHDF2 was demonstrated to increase its affinity for binding to M6A-modified RNA, which can promote the degradation of certain RNAs [24]. Mature mRNA still retains M6A, and mRNA translation is regulated through various mechanisms based on the position of M6A in mRNA. Research shows that YTHDF1 selectively recognizes the M6A modification site of the 3′ UTR and not only combines the methylated transcript with the ribosome but also recruits the eukaryotic initiation factor 3 (eIF3) of the translation initiation factor complex to interact with it, thus significantly improving the translation efficiency [25]. Interestingly, METTL3 in the cytoplasm can also serve as a reader for the translation of M6A-modified mRNA. If the methylated adenosine is located at the 5′ UTR, the binding of eukaryotic cell translation initiation factor 3 (eIF3) will recruit the 43S translation initiation complex to promote independent translation. However, if M6A is located in the coding region, it can slow down translation elongation efficiency by inhibiting the interaction between tRNA and transcripts [26].

### 2.2. Involvement in the Processing of microRNAs

MicroRNA is a type of evolutionarily conserved noncoding small molecule RNA, generally between 21–23 nucleotides in length, that has critical functions, such as regulating various biological processes by posttranscriptionally controlling the targeted genes’ expression [27,28,29]. Because the recognition sites of M6A modification and miRNAs binding on mRNA molecules are mainly concentrated near the 3’UTR and termination codon, M6A modification may coordinate with miRNA to regulate gene expression of mRNAs [30]. Studies have shown a strong positive correlation between the number of M6A sites and the binding of miRNAs and RNA-binding proteins (RBPs) to mRNAs, implying that M6A-modified mRNA is more easily targeted by miRNAs and RBPs [31]. Dgcr8 is a critical component of the microprocessor complex for the biogenesis of miRNAs, with the function of helping to cleave the primary miRNAs (pri-miRNAs) into pre-miRNAs to initiate miRNA production [32]. Alarcon et al. found that Dgcr8 can recognize, and label pre-miRNAs methylated by METTL3. In addition, the absence of METTL3 reduces the binding of DGCR8 microprocessor complex subunit (Dgcr8) with pre-miRNAs, leading to a global decrease in mature miRNAs and an accumulation of unprocessed pre-miRNAs [33]. In in vitro processing reactions, the effectiveness of M6A labeling in promoting pre-miRNA processing was confirmed.

### 2.3. Influence on the Structure and Function of LncRNA

M6A modification can affect lncRNAs in multiple ways. First, M6A modification can affect the structure and subsequently the function of lncRNAs. It was reported that M6A modification within nucleotides 2556–2587 of the lncRNA MALAT1 can induce a local structural change that increases the accessibility of the U5 tract for the recognition and binding of HNRNPC. The regulation of RNA–protein interactions through M6A-dependent RNA structural remodeling is termed the “M6A switch” [34]. There are 39,060 m6A switches identified among HNRNPC binding sites, and a reduction in global M6A decreases HNRNPC binding at 2798 high-confidence M6A switches. Importantly, m6A-switch-regulated HNRNPC binding affects the alternative splicing and abundance of target mRNAs [34]. Second, M6A modification is involved in the function of competing endogenous RNAs (ceRNAs). It was reported that lncRNA-PACERR induces tumorigenicity in pancreatic ductal cancer macrophages by interacting with miR-671-3P and the M6A reader IGF2BP2. LncRNA-PACERR binds to IGF2BP2, enhancing the stability of KLF12 and c-myc in the cytoplasm in an M6A-dependent manner [35]. Finally, M6A modification affects the RNA–RNA interactions of lncRNAs. METTL3-mediated M6A modification plays a role in linc1281-mediated RNA–RNA interactions, which are necessary for regulating pluripotency-related let-7 family microRNAs (miRNAs) and affecting mESC differentiation [4].

## 3. M6A RNA Methylation in Radiation-Induced Cellular Responses and Tissue Reactions

Radiation tissue reactions are based on cellular responses, referring to the detriment arising from noncancer effects of radiation on health, which were previously called ‘deterministic effects’ [36,37,38]. The manifestations of tissue damage and reactions vary from one tissue to another depending not only on irradiation doses but also on the cellular composition, proliferation rate, and intrinsic mechanisms of responses to radiation, which may be highly tissue specific. Recently, an increasing number of reports have demonstrated that M6A methylation plays an important role in regulating tissue reactions by affecting cellular responses to irradiation.

### 3.1. Impacts of Radiation on RNA M6A Methylation

The impacts of radiation on RNA M6A methylation may fluctuate across various types of radiation. Xiang et al. reported that increased RNA M6A methylation occurs on numerous mRNAs as soon as 2 min after nonionizing ultraviolet radiation (UV) irradiation, which is regulated by METTL3 and demethylase FTO [39]. It was confirmed that M6A methylase METTL3 is required for the immediate localization of DNA polymerase κ (Pol κ), a critical component in the nucleotide excision repair pathway of UV-induced DNA damage, to the site of DNA damage to fulfill its DNA repair function. In contrast, Yang et al. demonstrated that the “writing enzyme” METTL14 experienced a decline in *human* epidermal cells HaCaT and NHEK following ultraviolet B (UVB) irradiation, impeding global genome repair (GGR) [40]. Zhang et al. reported that METTL3-mediated M6A methylation directly participates in the process of X-ray ionizing radiation-induced DNA double-strand breaks repair in U2OS cells [41]. After being phosphorylated by ATM, METTL3 localizes to the DNA damage sites to catalyze M6A modification of RNAs, resulting in the accumulation of DNA-RNA hybrids at DSBs to facilitate the homologous recombination-mediated repair of DSBs. The impacts of ionizing radiation on the methylation level of RNA M6A are radiation dose- and post-IR time-dependents. A recent study showed that IR-induced alterations on M6A methylation level in a series of transcripts exhibited dose- and time-dependent effects in *mice*, *macaques*, *human* umbilical vein endothelial cells, and *human* peripheral blood cells [42,43]. M6A levels of Ncoa4, Ate1, and Fgt22 increased in a dose- and time-dependent manner after exposure to γ-rays and X-rays at doses ranging from 0.2 Gy to 6.5 Gy and post-IR times of 1 to 24 days in *mice* and *macaques* or *human* peripheral blood cells. These dose-dependent alterations of M6A methylation provide potential biomarkers for IR exposure. However, it is worth noting that radiation-induced alterations of RNA M6A methylation may be radiation type- and dose-specific, especially in in vitro cultured cell lines. Although a remarkably increased M6A level in RNAs was induced in U2OS cells by UV irradiation, no induction of M6A was observed by 10 Gy of γ-ray IR [39]. Our team’s latest research indicated that the overall M6A level of RNAs decreased after 2 h and until 12 h after 4Gy γ-ray irradiation in both HeLa and A549 cells [44], which is due to the depressed expression of METTL3 by IR. Transcriptome-wide M6A-seq and RNA-seq assays also indicate that the ratio of manuscripts with decreased M6A levels is higher than that of increased M6A levels.

### 3.2. M6A Methylation in Radiation-Induced DNA Damage Response to Determine Cellular Radiosensitivity

Wu et al. found that irradiation of *human* pharyngeal cancer cells can enhance cellular radioresistance by targeting the caspase1 pathway after affecting the expression of METTL3-mediated M6A modification and regulating cirCUX1 mRNA [45]. Another study showed that irradiation of pancreatic cancer cells MIA and PaCa-2 in combination with METTL3 knockdown can further enhance cellular radiosensitivity through MAPK cascades, ubiquitin-dependent processes, RNA splicing, and regulation of cellular processes [46]. Additionally, Shi et al. reported that the expression of circRNF13 is governed by METTL3/YTHDF2 controlling M6A methylation during irradiation, and circRNF13 mediates radioresistance of cervical cancer cells by increasing the stability of CXCL1 [47]. Consequently, different tissues harbor distinct genomes, transcriptomes, and immune characteristics, resulting in divergent mechanisms of M6A modification regulation following radiation exposure.

Multiple studies have demonstrated the involvement of RNA M6A methylation in the repair mechanism of radiation-induced DNA damage. When HEK293T and U2OS cells are exposed to UV irradiation, the TonEBP-METTL3-M6A RNA methylation pathway recruits RNaseH1 to damaged DNA, facilitating repair, decomposing R-loops, and subsequently enhancing cell survival rates [48]. Furthermore, the METTL3-METTL14 methyltransferase complex and YTHDC1 can be mobilized to DNA damage sites following UV and X-ray exposure [49]. Additionally, N6-methyladenine (N6mA) reduces the misincorporation of 8-oxo-guanine (8-oxoG) opposite to N6mA by DNA repair polymerases, thereby mitigating improper error DNA damage repair. Upon exposure to local laser microirradiation, the METTL16 enzyme is recruited to DNA damage sites, methylating nearby small RNAs (both snRNAs and snoRNAs) to mount a stress response to DNA damage [50]. As a key regulator of gene expression in cells, M6A plays a significant role in programmed cell death or cycle regulation post-radiation. In nasopharyngeal carcinoma radiotherapy, the demethyltransferase FTO promotes the expression of deubiquitylase (OTUB1) expression to counter ferroptosis and bolster the nasopharyngeal carcinoma radioresistance [51]. Additionally, METTL3 enables nasopharyngeal carcinoma cells to resist apoptosis after radiation by mediating the SUCLG2-AS1/CTCF/SOX2 axis [52]. Polo-like kinase 1 (PLK1) serves as a pivotal prognostic regulator in pancreatic cancer patients. Studies have revealed the crucial role of M6A methylation of PLK1 in maintaining the cell cycle post-radiation in pancreatic cancer cells [53]. Specifically, FTO demethylates PLK1 3’UTR, leading to reduced PLK1 expression at the G2/M phase, inducing mitotic abnormal replication pressure, and ultimately increasing cell death with a rise in the S-G2-M phase.

### 3.3. M6A Methylation in Radiation-Induced Injury of IR-Sensitive Hematopoietic Tissue

The hematopoietic system stands out as the most vulnerable to radiation-induced injury. Throughout irradiation, the proliferation or augmentation of myeloid-derived suppressor cells (MDSCs) impacts the tumor microenvironment, culminating in resistance to radiotherapy. Substantial evidence currently supports the regulatory role of M6A methylation in reshaping the tumor microenvironment and immune landscape influenced by radiation, thereby enhancing the efficacy of radiotherapy. YTHDF2, serving as a reader of M6A methylations, experiences upregulation in mice post-IR, triggering the activation of the NF-kB signaling pathway, elevating IL-10 in MDSCs, and instigating a transformation in the immune environment. However, the deletion of the YTHDF2 gene augments antitumor immunity by impeding MDSC differentiation, invasion, and migration, thus circumventing radiotherapy resistance [54]. Human hematopoietic stem/progenitor cells (HSPCs) swiftly and transiently trigger an increase in reactive oxygen species (ROS) upon exposure to low dose ionizing radiation, leading to enduring hematopoietic impairments [55]. Yu et al. uncovered that ROS fosters ALKBH5 SUMOylation by activating the ERK/JNK signaling pathway, resulting in METTL3 upregulation and localization to DNA damage sites, promoting RNA methylation related to DNA damage repair, and safeguarding HSPCs from ROS-induced DNA damage [56]. Additionally, Zhang et al. observed that alterations in the bone marrow-induced hematopoietic system in mice following 4Gy γ-irradiation were linked to time-dependent epitranscriptome-wide M6A methylome and transcriptome changes. Nevertheless, the silencing of M6A demethylases FTO and ALKBH5 could mitigate radiation-induced hematopoietic damage [57].

### 3.4. M6A Methylation in Radiation-Induced Toxins of Organs

Radiotherapy (RT) is a common treatment for cancer patients. However, RT-induced organ injury presents a significant challenge for clinical treatment. Recent studies have shown that M6A methylation can participate in the regulation of radiation-induced tissue injury. Professor Zhaochong Zeng’s team has found that irradiation-induced METTL3-dependent M6A modification increases, activating the TEAD1-STING-NLRP3 signaling pathway to promote radiation-induced liver injury (RILD) [58]. Additionally, they have discovered that ALKBH5, acting as an “eraser” of M6A methylation, can regulate the HMGB1-STING-IRF3 axis to prevent RILD in irradiated liver tissue [59]. M6A modification may also be involved in radiation-induced lung injury (RILI). A recent study reported that after irradiation of lung tissue, METTL3-YTHDF2 regulates forkhead box O1 (FOXO1) mRNA stability through M6A modification, activating the EPK and AKT signaling pathways, ultimately leading to EMT [60]. Furthermore, Zhao et al. found that Zinc finger and BTB domain-containing protein 7B (Zbtb7b) reduce radiation-induced IL-6 production in the lung by inhibiting M6A modification of IL-6 mRNA and nuclear transport, which provides a new target for the treatment of radiation pneumonitis [61]. Surprisingly, it was reported that M6A modification could be regulated by biomass-derived carbon dots (CDs) to reduce radiation-induced bone injury [62]. The specific mechanism involves an increase in bone marrow mesenchymal stem cells (BMSCs) in rats after irradiation, dependent on M6A modification to promote the degradation of CAP-GLY domain containing linker protein 3 (CLIP3) mRNA, downregulating CLIP3 expression, and eventually leading to relief from radiation-induced bone injury.

## 4. M6A Methylation in RNA during Cellular Responses to Cancer Radiotherapy

During the radiotherapy process, tumor cells can undergo a series of complex biological responses and changes, consequently resulting in cancer cell death and tumor suppression. However, when cancer cells develop resistance to ionizing radiation or severe side effects are induced in normal tissues, the effectiveness of radiation therapy is greatly compromised [16,36,63]. There are different molecular aspects of the mechanisms for the radioresistance of tumors, including intrinsic genetic or epigenetic changes in cancer cells, the microenvironment of tumors, the existence of a small number of radioresistant cancer stem cells, etc. Recently, a series of reports have suggested that M6A modifications are critical mechanistic responses of cells to irradiation and contribute to clinical outcomes of cancer radiotherapy [64,65]. Once M6A regulatory factors are dysregulated, they change the sensitivity of cancer cells as well as tumors to radiotherapy. In addition, the components of the M6A modification process are commonly dysregulated on expression level or gene sequence alteration in cancer cells, which can largely influence cancer cells’ radiosensitivity.

### 4.1. M6A Writers Are Involved in the Response of Cancer Cells to Radiotherapy

METTL3 has been reported to have a favourable effect on tumour growth and is a risk factor for cancer prognosis in various tumour [66,67,68,69]. Visvanathan et al. reported a higher level of M6A modification glioma stem-like cells (GSCs) and METTL3-dependent M6A modification is crucial for GSC maintenance [45]. When METTL3 is silent, the sensitivity of GSCs to γ-rays is enhanced, and the DNA double-strand breaks repair efficiency is affected as indicated by the accumulation of DSBs biomarker γ-H2AX. After irradiation, the METTL3 and M6A modification in GSC increase, causing Human antigen R (HuR), an essential regulator of RNA metabolism, to bind more effectively to M6A -modified RNA, leading to an enhanced stability of Sex-determining region Y-box2 (SOX2), an important transcriptional regulator in pluripotent stem cells including cancer stem cells. METTL3 deficiency decreases SOX2 mRNA stability and protein level. The METTL3-SOX2 axis increases the activity of DNA DSBs homologous recombination repair pathway, promoting tumour radioresistance. It was reported that METTL3 mediates M6A modification of SUCLG2-AS1 transcript, a super enhancer associated lncRNA (SE-lncRNA), which is then recognized and stabilized by IGF2BP3. Consequently, SUCLG2-AS1 binds to the SE region of SOX2 to regulate SOX2 transcription, contributing to metastasis and radioresistance of nasopharyngeal carcinoma [52]. There is another report indicates that METTL3-mediated M6A modification involves in the prognosis and IR-induced alteration of cell cycle progressing of pancreatic cancer cells by targeting PLK1 mRNA 3’UTR [53]. In this report, IGF2BP2 was uncovered to bind to M6A of PLK1 3’UTR, leading to upregulation of PLK1 expression. Demethylation of this site increases mitotic catastrophe and radiosensitivity of cancer cells.

Lysophosphatidic acid receptor 5 (LPAR5) is considered a prognosis-related gene for pan cancers, and abnormal high-level expression of LPAR5 confers IR-induced epithelial-to-mesenchymal transition (EMT) and radioresistance to cancer cells [17,70]. Recently, our group identified that the A1881 in LPAR5 mRNA 3′UTR is an M6A-modified site mediated by METTL3, which determines the stability of LPAR5 mRNA [44]. METTL3 was depressed in HeLa and A549 cells during the first hours of post-irradiation of 4 Gy γ-ray via the pathway of PARP1-related chromosomal accessibility of the METTL3 promoter region. PARP1 inhibitors can also depress METTL3 expression, resulting in a decreased M6A level of LPAR5 mRNA A1881, consequently leading to decreased stability of LPAR5 mRNA and exhibiting a synergistic effect of tumor growth suppression with radiotherapy [44]. The roles and mechanisms of M6A writers in the regulation of cancer radiosensitivity are outlined in Figure 2.

In addition, other methyltransferases can also play a role in the radiotherapy of cancer cells. For example, Wang et al. found that neuropilin 1 (NRP1) downregulates Bcl2 through the M6A-dependent methyltransferase WTAP, thereby reducing IR-induced apoptosis in breast cancer cells [71]. Yang et al. suggest that METTL14 relies on M6A modification to promote damage specific DNA binding protein (DDB) translation and inhibit UVB radiation-induced skin cancer formation [40]. UVB radiation downregulates METTL14 protein expression through NBR1 autophagy cargo receptor (NBR1)-dependent selective autophagy, further reducing global genome repair (GGR) and translation of DDB2 transcripts and inducing the development of skin cancer. These findings suggest that METTL14 is a target for selective autophagy. It is a key epitope transcription mechanism that regulates GGR and inhibits UVB-induced skin tumorigenesis.

### 4.2. M6A Erasers Are Involved in the Response of Cancer Cells to Radiotherapy

Several demethyltransferases of M6A were also revealed to directly affect the radiotherapy effect of cancers, as shown in Figure 2. The toxic effect on normal tissues is a critical restrictive factor to cancer radiotherapy. Chen et al. reported the role of ALKBH5-mediated demethylation of RNA M6A in radiation-induced liver disease (RILD) [59]. They found that when normal liver tissue is irradiated by X-rays, ALKBH5 is recruited and demethylates M6A residues of high-mobility group protein 1 (HMGB1) mRNA 3’UTR, leading to the activation of STING-IRF3 signaling and increased liver cell apoptosis induction. ALKBH5 knock-out can attenuate the STING signal mediated by HMGB1 and reduce liver inflammation in vivo. Kowalski-Chauvel et al. reported that ALKBH5 can promote radiation resistance and the invasion ability of glioma stem cells [72]. The overexpression of ALKBH5 in glioma-associated mesenchymal stem cells (GBMSCs) promotes its radiation resistance by controlling homologous repair. In contrast, knocking down ALKBH5 in GBMSCs reduces the basal expression of RAD51 and IR-induced expression of RAD51, BRCA2, BRIP1, EXO1, and XRCC2 while promoting the IR-induced expression of checkpoint kinase 1 (CHK1), further inhibiting the ability of GBMSCs to repair DNA damage and leading to cell radiosensitization. In addition, silencing the M6A demethylases FTO and ALKBH5 can also alleviate radiation-induced hematopoietic damage [57]. It was reported that FTO is significantly upregulated in radioresistant nasopharyngeal carcinoma (NPC) tissues. Mechanically, FTO, as a M6A demethylase, clears the M6A modification of the OTUB1 transcripts and consequently enhances the expression of OTUB1, thereby promoting OTUB1-mediated antiferroptosis and NPC radioresistance [51].

### 4.3. M6A Readers Involved in Cancer Radiotherapy

M6A modification is related to the local control and metastasis of tumors after radiotherapy. It was reported that the absence of M6A reader YTHDF2 alters the differentiation of immunosuppressive myeloid-derived suppressor cells (MDSCs), inhibits the invasion of MDSCs, thus enhancing antitumor immunity and overcoming tumor radiation resistance [54]. In a mouse model, IR induced the expression of YTHDF2 by activating NF-kB, leading to downregulated expression of its direct targets, adrenoceptor beta 2 (ADRB2), meteorin like glial cell differentiation regulator (Metrnl), and sphingomyelin phosphodiesterase acid like 3B (Smpdl3b), thereby negatively regulating NF-kB signal transduction. The YTHDF2 inhibitor DC-Y13-27 enhanced the antitumor effect of a combination of radiation therapy and radioimmunotherapy in a manner similar to YTHDF2 deficiency. He et al. also found that the M6A reader YTHDC2 promotes radiotherapy resistance in nasopharyngeal carcinoma by activating the IGF1R/AKT/S6 signal axis [73]. M6A-reading protein IGF2BPs was also reported to be associated with the radioresistance of lung adenocarcinoma cells [74]. Irradiation upregulates VANGL planar cell polarity protein 1 (VANGL1) mRNA M6A level and expression, while IGF2BP2/3 deficiency downregulates VANGL1 mRNA stability and expression. Increased VANGL1 can augment B-Raf proto-oncogene, serine/threonine kinase (BRAF) protein stability and expression of its downstream DNA repair proteins TP53BP1 and RAD51. Yang et al. found that, compared to normal *human* skin, YTHDC2 was highly expressed in *human* skin squamous cell carcinoma (CSCC). When YTHDC2 was knocked down in CSCC cells, the expression of the PRC2 component SUZ12 and the levels of the histone modification H3K27me3 decreased. At the same time, the expression of phosphatase and tensin homolog (PTEN) increased, and the repair of UVB-induced DNA damage was enhanced. Interestingly, METTL14 knockdown reversed the inhibitory effect of YTHDC2 on DNA repair, while the inhibitory effect of M6A eraser FTO was similar to that of YTHDC2 [40]. The effects of M6A readers on cancer radioresistance are illustrated in Figure 2.

## 5. Conclusions and Perspectives

The M6A modification of RNA plays a crucial regulatory role in tumor occurrence, development, and radiation sensitivity. Meanwhile, M6A modification is generally believed to be a dynamically balanced regulatory process, where M6A regulatory factors (writers, erasers, and readers) play a reversible regulatory role in the modification of M6A to maintain a certain steady state. To further explore the key effects of M6A in tumor radiotherapy, we believe that further research is needed on the balance of M6A regulatory factors in tumor cells and adjacent tissues. Research in this area can not only propose new strategies to address the sensitivity of tumor radiotherapy but also provide effective references for protecting normal body tissues during the radiotherapy process.

Second, we believe that precise detection of M6A modification levels is crucial for research in this field, and there is currently a lack of precise detection methods that can directly detect M6A modification. M6A seq or MERIP seq used anti-M6A to enrich RNA fragments containing M6A, producing maps with a resolution of 100–200 nucleotides (NT) [30,75]. In addition, this antibody-based method improved versions of PA-M6A-seq [76], miCLIP [77], and M6A laic-seq [78] and made it possible to conduct extensive research on M6A and its biological functions. However, antibody-based methods have several significant limitations, including low resolution, lack of stoichiometric information, the requirement for a large amount of input materials (such as >20 mg total RNA), and limited ability to compare M6A methylation under different conditions [79]. A quantitative M6A-SAC-seq method proposed by Lulu Hu and the M6A spectrum detected by this technology, which needs only ~30 ng of input RNA (from 300 ng total RNA), can obtain whole-transcriptome M6A stoichiometry maps in various cell differentiation, early development, neuronal signaling, and clinical samples. Our future research will be focused on in situ detection methods for M6A modification of RNA with the advancement of science and technology. According to the proportion of RNA types, the proportion of mRNA is very small, and the majority is noncoding RNA. Thus far, reports on the impact of M6A modification on RNA structure and function have been very limited. The structural function of noncoding RNA has a significant impact on its regulatory mechanisms, such as the microRNA-like function of snoRNA [80,81], and so on. It is currently known that RNA modification has an important impact on the tertiary structure of RNA. By combining in situ M6A modification detection with nuclear magnetic resonance analysis [82] of the three-dimensional structure of RNA, we can discover more about the regulatory mechanism of M6A.

In summary, M6A modification in tumors has been receiving more research attention in recent years. It has not only revealed a new epigenetic biology in tumor cells but also provided new insights into the molecular mechanisms of tumor occurrence, prognosis, radiation therapy, drug resistance, etc. It is even possible to develop new antitumor drugs that play an enormous role in the treatment of various types of cancer.

## Figures and Tables

**Figure 1 ijms-25-02597-f001:**
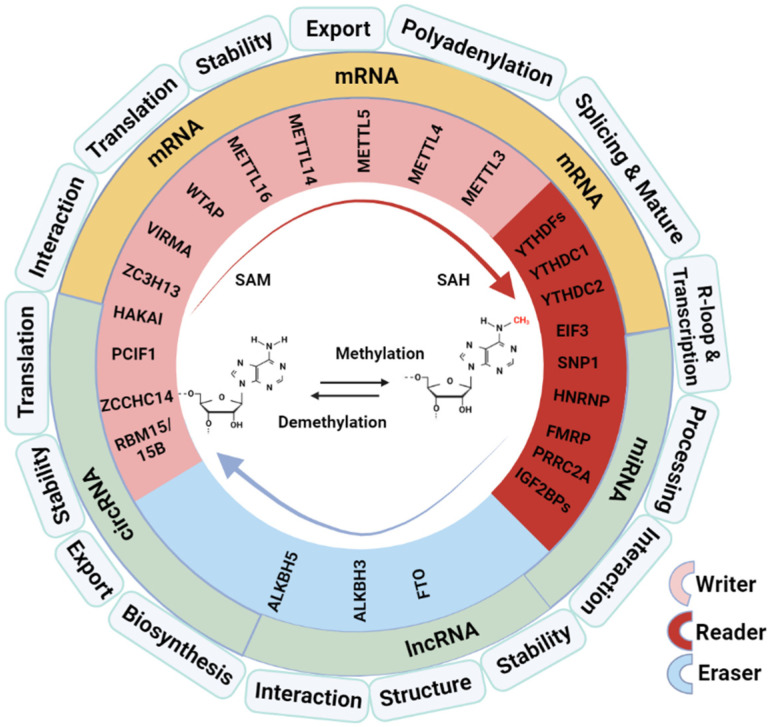
Overview of M6A modifiers and regulation on RNA processing and functions. SAM: methyl donor S-adenosylmethionine; SAH: S-adenosyl-L-homocysteine. The graphical was created with Biorender (https://app.biorender.com) on 10 January 2024 and a license was granted (SF26BNDZ9M).

**Figure 2 ijms-25-02597-f002:**
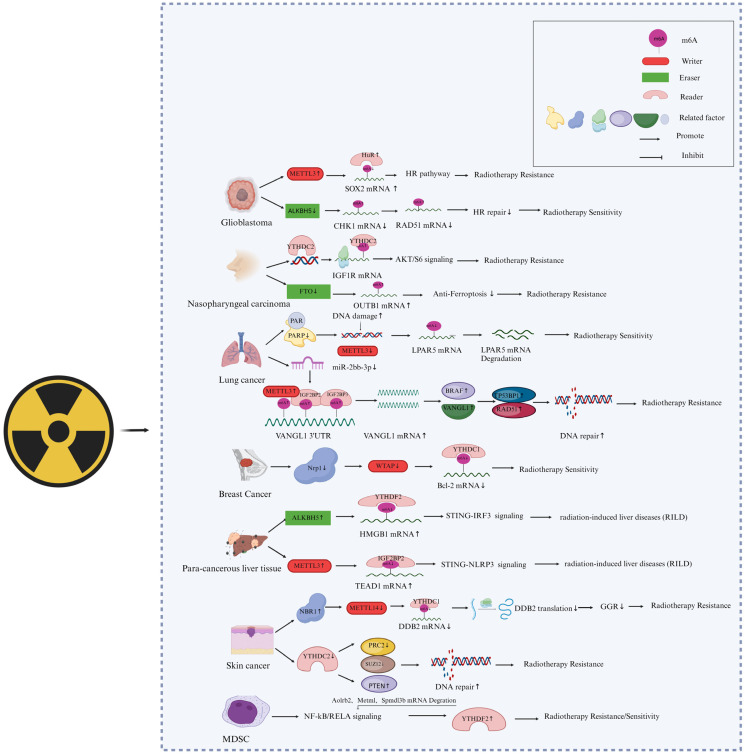
Involvement of M6A “writers”, “erasers”, and “readers” in the responses of cancer cells to radiotherapy. The graphical was created and revised with Biorender (https://www.biorender.com/, accessed on 20 February 2024), and a license was granted (UL26HEBPMU).

## Data Availability

Not applicable.

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
