# Peer review of "RNA N6-Methyladenosine Modification in DNA Damage Response and Cancer Radiotherapy"

_ijms, 2024, doi:10.3390/ijms25052597_

Round 1
Reviewer 1 Report
Comments and Suggestions for Authors
This manuscript reviews current insights about RNA N6-methyladenosine (M6A) modification and its association with ionizing radiation.
1. In chapter 3.1, How M6A methylation is regulated and M6A methyltransferases are activated upon irradiation? Is METTL3 phosphorylation by ATM only associated with localization to the DNA damage site upon X-ray irradiation?
2. As shown in Figure 2-4, M6A modification plays multiple roles in cancer radiotherapy. However, it looks like each effect depends on specific pathway for each cancer type (or each paper cited). Simplifying the figure will make more better understanding if possible.
3. Related to chapter 4, Does each M6A writers, readers and erasers specifically recognize certain groups of RNAs, or determine the trend of mRNA degradation and protein expression upon irradiation?
4. Manuscript editing is not sufficient. There are so many typos (for example, line 12, 27, 33, 69, 75, 83, 103, ...) inconsistencies in spacing (for example, line 28, 33-35, ...) and citation style (for example, first name-last name and last name-first name is coexisting). Please check carefully and use manuscript editing service if necessary.

Comments on the Quality of English Language-
Author Response
- In chapter 3.1, How M6A methylation is regulated and M6A methyltransferases are activated upon irradiation? Is METTL3 phosphorylation by ATM only associated with localization to the DNA damage site upon X-ray irradiation?
Response: The effect of ionizing radiation on the expression of m6A regulators results in alterations in the overall m6A modification level in cells or tissues. However, there are few reports on the specific mechanism by which ionizing radiation regulates m6A regulators. The latest research in our laboratory showed that PARP1, an important regulator of DNA damage repair, can regulate the transcription of METTL3 and the m6A methylation of poly(A)+ RNA by regulating the chromatin accessibility of HeLa and A549 cells exposed to 4 Gy of gamma rays to cope with radiation-induced DNA damage. At present, only one study has reported that METTL3 phosphorylation by ATM is associated with localization to a DNA damage site only upon X-ray irradiation, and two other studies have reported that METTL3 phosphorylation by ATM is related to the chemical bleomycin (https://doi.org/10.1016/j.canlet.2023). 216092) and PP2Ac α (https://doi.org/10.1155/2021/1015293)
- As shown in Figure 2-4, M6A modification plays multiple roles in cancer radiotherapy. However, it looks like each effect depends on specific pathway for each cancer type (or each paper cited). Simplifying the figure will make more better understanding if possible.
Response: Thank you for proposing that we agree with you that “simplifying the figure”. Therefore, we modified the figure according to the effect of M6A modification on different cancer types.
- Related to chapter 4, Does each M6A writers, readers and erasers specifically recognize certain groups of RNAs, or determine the trend of mRNA degradation and protein expression upon irradiation?
Response: According to the current reports, m6A modification has specificity only for recognized sequences (motifs, RRACH) on RNA sequences and has no specificity for RNA types. Because m6A modification results in a dynamic and reversible equilibrium state, the functional effects of m6A on various types of RNA also vary according to the specific modifying enzymes. The three modifying enzymes at the same site on the same RNA are not singular. However, it has specificity for modifying sites. For example, the functions of the 3'UTR and exon regions are different.
- Manuscript editing is not sufficient. There are so many typos (for example, line 12, 27, 33, 69, 75, 83, 103, ...) inconsistencies in spacing (for example, line 28, 33-35, ...) and citation style (for example, first name-last name and last name-first name is coexisting). Please check carefully and use manuscript editing service if necessary.
Response: In our resubmitted manuscript, the formatting and quotation style errors have been corrected. Thank you for your correction.

Reviewer 2 Report
Comments and Suggestions for Authors
The presented manuscript is a review of the RNA N6-methyladenosine Modification in DNA Damage Response and Cancer Radiotherapy. It is an interesting topic. However, I have a few comments.
1. Chapters 1. Introduction, and 2. The role of M6A modification in RNA metabolism and functions (lines 22-186) should be shortened because they do not present new information that has been better presented in other publications (e.g. https://doi.org/10.1002/cac2.12458, https://doi.org/10.3390/ijms19020555).
2. The relevant part of the work relating to the title does not sufficiently present the described mechanisms in the context of specific types of cancer which has also been presented in other publications (e.g. https://doi.org/10.1007/s12672-023-00759-3, https://doi.org/10.3390/ijms19020555).
3. The entire text of the work was prepared carelessly. There are numerous editorial errors in the text (e.g. extra spaces, missing spaces).
4. Author 1, A.B.; Author 2, C.D. Title of the article. Abbreviated Journal Name Year, Volume, page range.
Author Response
- Chapters 1. Introduction, and 2. The role of M6A modification in RNA metabolism and functions (lines 22-186) should be shortened because they do not present new information that has been better presented in other publications (e.g., https://doi.org/10.1002/cac2.12458, https://doi.org/10.3390/ijms19020555).
Response: Thank you for your suggestion and we have shortened and revised the contents of these two sections. For example, this paragraph “The molecular compositions of RNA M6A modification working system include the “writers” (adenosine methyltransferases), “readers” (RNA binding proteins) and “erasers” (demethylases)….. impacts mRNA stasbility.” has been largely shorten.
- The relevant part of the work relating to the title does not sufficiently present the described mechanisms in the context of specific types of cancer which has also been presented in other publications (e.g. https://doi.org/10.1007/s12672-023-00759-3, https://doi.org/10.3390/ijms19020555).
Response: Based on your suggestions, we have supplemented and modified the content of this section.
- The entire text of the work was prepared carelessly. There are numerous editorial errors in the text (e.g. extra spaces, missing spaces).
Response: In our resubmitted manuscript, the editorial errors have been extensively corrected.
- Author 1, A.B.; Author 2, C.D. Title of the article. Abbreviated Journal Name Year, Volume, page range.
Response: According to the citation style of IJMS journals, we have revised the references. Thank you for your reminder.

Round 2
Reviewer 2 Report
Comments and Suggestions for Authors
The Authors have satisfactorily responded to all my questions and made the necessary changes to the manuscript.